# Experimental reconstitution of chronic ER stress in the liver reveals feedback suppression of BiP mRNA expression

Javier A Gomez[1], D Thomas Rutkowski[2]*

[1]Graduate Program in Molecular and Cellular Biology, University of Iowa Carver College of Medicine, Iowa City, United States; [2]Departments of Anatomy and Cell Biology and Internal Medicine, University of Iowa Carver College of Medicine, Iowa City, United States

**Abstract** Endoplasmic reticulum (ER) stress is implicated in many chronic diseases, but very little is known about how the unfolded protein response (UPR) responds to persistent ER stress in vivo. Here, we experimentally reconstituted chronic ER stress in the mouse liver, using repeated injection of a low dose of the ER stressor tunicamycin. Paradoxically, this treatment led to feedback-mediated suppression of a select group of mRNAs, including those encoding the ER chaperones BiP and GRP94. This suppression was due to both silencing of the ATF6$\alpha$ pathway of UPR-dependent transcription and enhancement of mRNA degradation, possibly via regulated IRE1-dependent decay (RIDD). The suppression of mRNA encoding BiP was phenocopied by ectopic overexpression of BiP protein, and was also observed in obese mice. Our findings suggest that persistent cycles of UPR activation and deactivation create an altered, quasi-stable setpoint for UPR-dependent transcriptional regulation—an outcome that could be relevant to conditions such as metabolic syndrome.

*For correspondence: thomas-rutkowski@uiowa.edu

**Competing interests:** The authors declare that no competing interests exist.

## Introduction

The western obesity epidemic has exposed one of the consequences of modernity: the increasing prevalence of non-infectious chronic diseases that progress down increasingly irreversible paths over the course of years to decades. Diabetes, atherosclerosis, hypertension, steatohepatitis, and the like join neurodegenerative disorders, cancers, lung disease and other chronic diseases in driving morbidity and mortality in the western world (*Ezzati et al., 2002*). As a class, these diseases necessarily entail a *gradual* deterioration of cellular and organ function, rather than an acute collapse. Thus, treating or reversing them requires understanding how persistent but otherwise modest stimuli alter the activity of key cellular pathways.

One cellular pathway increasingly implicated in the progression of a number of chronic diseases is the Unfolded Protein Response (UPR). The UPR is activated by disruption of the protein folding capacity of the endoplasmic reticulum, otherwise known as 'ER stress' (*Walter and Ron, 2011*). Unremitted ER stress and/or a dysregulated UPR appear to contribute to hepatic steatosis and steatohepatitis (*Malhi and Kaufman, 2011*), atherosclerosis (*Zhou and Tabas, 2013*), colitis and inflammatory bowel disease (*Kaser et al., 2013*), hypertension (*Young and Davisson, 2015*), and many others. As an organelle that carries out several essential cellular processes (protein processing, calcium storage, lipogenesis, certain metabolic steps, etc.) and that is physically and functionally intertwined with many other critical cellular pathways, the ER is sensitive to a range of diverse stimuli. For example, ER stress is observed in the livers of obese mice (*Ozcan et al., 2004*), and this response has been attributed to an excess load of nascent client proteins in the organelle due to nutrient-

**eLife digest** Toxic chemicals, extreme temperatures and other abnormal environmental conditions can cause the cells in our bodies to become stressed. Several kinds of stresses overwhelm a compartment in the cell called the endoplasmic reticulum, which is critical for processing new proteins so that they can work correctly. Endoplasmic reticulum stress has been linked to long-term diseases such as diabetes, cancer and neurodegenerative diseases.

Most of what is known about how cells sense and respond to endoplasmic reticulum stress comes from studies on isolated cells that were subjected to harsh conditions that cells cannot tolerate for longer than a day or two. By contrast, little is known about how cells within whole organisms respond to milder but longer-lasting endoplasmic reticulum stress, which is closer to what occurs during disease.

To investigate this issue, Gomez and Rutkowski treated mice repeatedly with a chemical that causes mild endoplasmic reticulum stress in the liver. The cells exposed to this persistent stress responded differently to those exposed to severe short-term stress. Whereas short-term stress causes liver cells to turn on genes that help the endoplasmic reticulum to process proteins more efficiently, long-term stress causes cells to turn off some of those genes.

Further investigation revealed that cells in the livers of obese mice show similar patterns of gene activity as cells exposed to long-term endoplasmic reticulum stress. The findings presented by Gomez and Rutkowski could therefore also help us to understand more about the liver problems that often occur during obesity and diabetes.

Further studies are now needed to examine exactly how long-lasting stress can shut off the cells' protective mechanisms. Future experiments could also investigate whether other types of cells and organs respond to long-term endoplasmic reticulum stress in the same way as cells in the liver.

mediated stimulation of mTOR activity (*Ozcan et al., 2008*). Other physiological stimuli that elicit ER stress include nutritional status and the activity of metabolic pathways (*Oyadomari et al., 2008*; *Tyra et al., 2012*; *Shao et al., 2014*), differentiation cues (*Iwakoshi et al., 2003*; *van Anken et al., 2003*; *Lee et al., 2005*), inflammatory signals (*Zhang et al., 2006*; *Hotamisligil, 2010*), and many others. While the UPR is capable of responding to excessive ER stress by initiating cell death cascades (*Sano and Reed, 2013*), chronic stress is instead likely to result not in cell death (at least not for most cells, most of the time), but instead in a persistent burden on ER function that must be accommodated by the UPR. And yet, for many chronic diseases in which ER stress is implicated, it is not clear whether disease results from a UPR that becomes progressively dysregulated and thus responds *inappropriately*; or whether the UPR becomes progressively neutered and simply becomes increasingly *unresponsive*.

Experimental manipulation of the UPR has sketched the framework of a canonical UPR that is initiated by three ER-resident proteins and that culminates in transcriptional augmentation of the ER protein processing capacity and other non-transcriptional mechanisms to alleviate ER load. The inositol-requiring enzyme 1α (IRE1α) pathway results in both the production of the XBP1 transcription factor (*Yoshida et al., 2001*; *Calfon et al., 2002*; *Lee et al., 2002*) and the degradation of ER-associated mRNAs in a process known as Regulated IRE1-Dependent Decay via the IRE1 endonuclease activity (*Hollien and Weissman, 2006*; *Hollien et al., 2009*). The PKR-like endoplasmic reticulum kinase (PERK) stimulates production of the ATF4 transcription factor (*Harding et al., 2000*) and also transiently reduces ER protein load by phosphorylation of the translation initiation factor eIF2α (*Shi et al., 1998*; *Harding et al., 1999*). ATF6 (comprising both α and β paralogs) initiates the third pathway when ATF6 is itself cleaved by regulated intramembrane proteolysis in the Golgi, liberating a transcriptionally active N-terminal fragment (*Ye et al., 2000*). These pathways are all robustly activated by acute exposure of cells to high doses of ER stress-inducing drugs, which is an effective tool for understanding what the system is capable of but not as effective for understanding how the cellular response to chronic stress differs from these canonical pathways.

Evidence for the involvement of ER stress in chronic diseases in general comes from two lines of investigation: detecting markers of ER stress in the affected tissues of human patients or mouse

models thereof, and observing disease-like phenotypes in mice with genetic ablations of UPR signaling pathways. Yet while studies of this sort provide *prima facie* evidence for UPR involvement, they do not address how the UPR becomes dysregulated during chronic disease, if in fact it even does. Further, animal models of chronic diseases such as those associated with obesity tend to influence innumerable cellular pathways and make it difficult to isolate the effects of ER stress per se from the many other confounding factors.

With these considerations in mind, we set out to address how the UPR responds to chronic stress in vivo. Following the example of our previous work in vitro (*Rutkowski et al., 2006*), here we describe how UPR signaling in the liver becomes altered during repeated exposure to stress, the mechanisms by which these alterations occur, and their potential relevance to liver dysfunction during obesity.

## Results

### The liver adapts to persistent ER stress

Our first step in experimentally reconstituting chronic ER stress was to identify conditions that were as specific as possible in perturbing ER function with minimal pleiotropic effects, and were sufficient to activate the UPR without causing overt toxicity. In doing so, we hoped to be able to understand how the UPR responds to chronic stress, divorced from the confounding influences of chronic diseases on other cellular pathways. Toward this end, we used tunicamycin (TM) to induce ER stress in vivo; TM blocks N-linked glycosylation, an ER-specific protein modification. It has few known off-target effects, most prominently targets the kidneys and the liver, and is not lethal in wild-type animals at even relatively high doses (*Foufelle and Fromenty, 2016*). Its effects on the liver can also be mimicked, albeit less readily, by other agents that elicit ER stress in that organ (*Rutkowski et al., 2008*; *Chikka et al., 2013*), further justifying its use here. Following an approach analogous to one we described previously in cultured cells (*Rutkowski et al., 2006*), we looked for a dose of TM that was capable of inducing the UPR, but that was significantly less toxic than the common experimental dose of 1 mg/kg—which is known to elicit hepatocellular death and inflammation (*Foufelle and Fromenty, 2016*). Using qRT-PCR to detect changes in expression of the UPR sentinel genes *Hspa5* (encoding BiP) and *Ddit3* (encoding CHOP), we found that doses of TM below 0.1 mg/kg were capable of eliciting attenuated UPR activation (*Figure 1A*).

Based on this finding, we chose 0.025 mg/kg as the dose most likely to be tolerated long-term by mice without leading to significant toxicity, and we used it as the basis for induction of chronic stress. As expected, acute treatment (8 hr) with this dose of TM was sufficient to lead to upregulation of both BiP and CHOP at the protein level (*Figure 1B*), although for CHOP this induction was much more modest than commonly elicited by higher doses of TM. To test the effects of chronic treatment with 0.025 mg/kg TM, we injected animals with this dose (or with vehicle) every day for five days (or longer), as shown in *Figure 1C*. Animals were sacrificed approximately 24 hr after the most recent dose (i.e., immediately before the next injection would have taken place). After sacrifice, livers were harvested and analyzed for molecular and histological markers of ER stress.

The predominant acute effect of TM on the liver, as revealed by hematoxylin and eosin staining, was the formation of intracellular vacuoles in hepatocytes (*Figure 1D*) that correspond to accumulated lipid droplets resulting from impaired lipid metabolism (*Rutkowski et al., 2008*). These vacuoles were abundant in animals administered a single dose of 0.025 mg/kg TM, but, remarkably, were absent by the end of the fifth day of challenge. Other than these changes, there were few signs of gross liver damage such as necrosis or fibrosis (data not shown). Thus, at least by these histological criteria, animals are capable of maintaining liver function in the face of ongoing stress. The apparent normalization of liver histology could be explained if animals became resistant to TM over time. However, persistent underglycosylation of the ER-resident glycoprotein TRAPα at day five confirmed that the drug remained active upon repeated dosing (*Figure 1E*).

In order to test whether ER stress persisted in the chronic condition despite the apparent normalization of liver histology, we examined expression of BiP and CHOP proteins. Immunoblotting showed that BiP upregulation persisted throughout the time-course (*Figure 1F*), as did upregulation of the ER chaperone GRP94 (not shown). In contrast, and consistent with the effects of chronic stress on cultured cells (*Rutkowski et al., 2006*), upregulation of the highly labile CHOP protein was lost

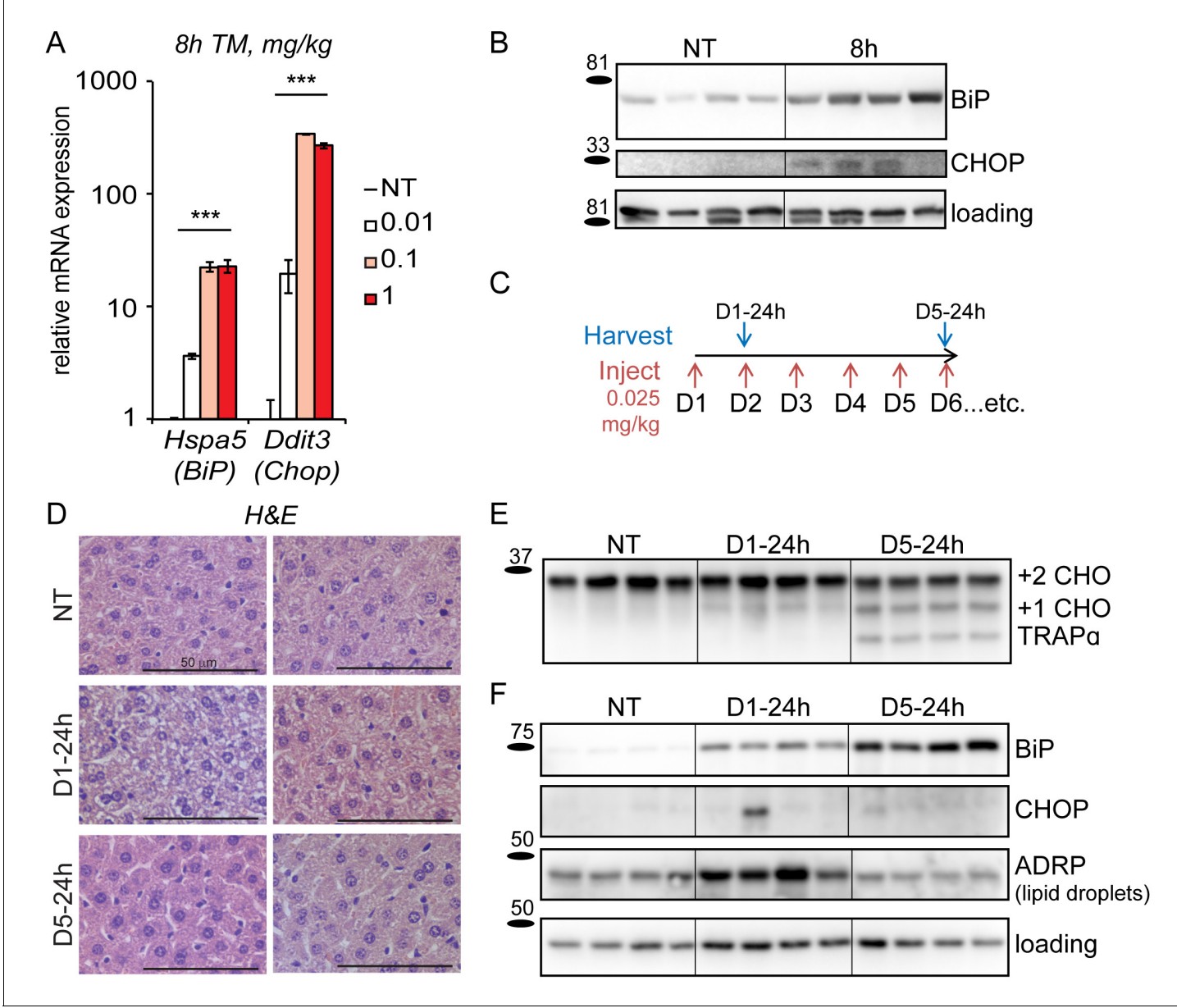

**Figure 1.** Mice adapt to repeated exposure to TM despite persistent stress. (A) Livers from wild-type mice collected 8 hr after injection with the indicated concentrations of TM (in mg/kg) were probed by qRT-PCR for expression of *Hspa5* (encoding BiP) and *Ddit3* (encoding CHOP). n = 2–3 animals/group. Significance was determined by one-way ANOVA. Here and elsewhere, ***, p<0.001; **, p<0.01; *, p<0.05.; NS = not significant (B) Livers were harvested after treatment for 8 hr with vehicle or 0.025 mg/kg TM and protein lysates were probed by immunoblot to detect BiP or CHOP. Calnexin was used as a loading control. For all blots and gels in this paper, each lane represents a separate animal. Hairlines are used for visual clarity only. (C) Chronic stress treatment schematic. Mice were weighed and then injected daily with 0.025 mg/kg TM (red arrows); liver samples were collected at the indicated times (blue arrows) for downstream analysis. The naming convention is as follows: 'D' indicates the number of daily injections received while 'h' indicates the time of tissue collection after the last injection. Here and elsewhere, not treated animals (NT) received injections of vehicle at the same times that D5-24h mice received TM. (D) Formalin-fixed liver sections from animals treated as in (C) were collected, fixed, and stained by H and E at the indicated times. For each condition, representative images from two separate mice are shown. Note the extensive cytoplasmic vacuolization in the D1-24h animals. (E, F) Protein lysates from animals treated as in (C) were isolated and probed for expression of the ER-resident glycoprotein TRAPα (E) and BiP, CHOP, ADRP, and α-actin (loading control) (F).

The following source data is available for figure 1:

**Source data 1.** Contains raw and transformed Ct values for qRT-PCR experiment in *Figure 1A*.

even by D1-24h, and remained mostly undetectable in animals treated for 5 days (*Figure 1F*). We also monitored the lipid droplet marker protein ADRP, the expression of which correlates with lipid droplet content. Consistent with histological data (*Figure 1D*), ADRP expression—and therefore intracellular lipid accumulation—was elevated in animals treated for one day, but returned to basal levels in animals treated for five days (*Figure 1F*). Of note, we also performed a similar experiment in which the chronic treatment was extended to 15 days. Results were similar to the 5 day treatment (not shown; see also *Figure 2A*).

Taken together, these data indicate that chronic TM treatment caused persistent ER stress in the liver without permanently perturbing liver anatomy. They suggest that, as with cultured cells, hepatocytes in vivo are capable of adapting to long-term disruption of ER function if the stress is

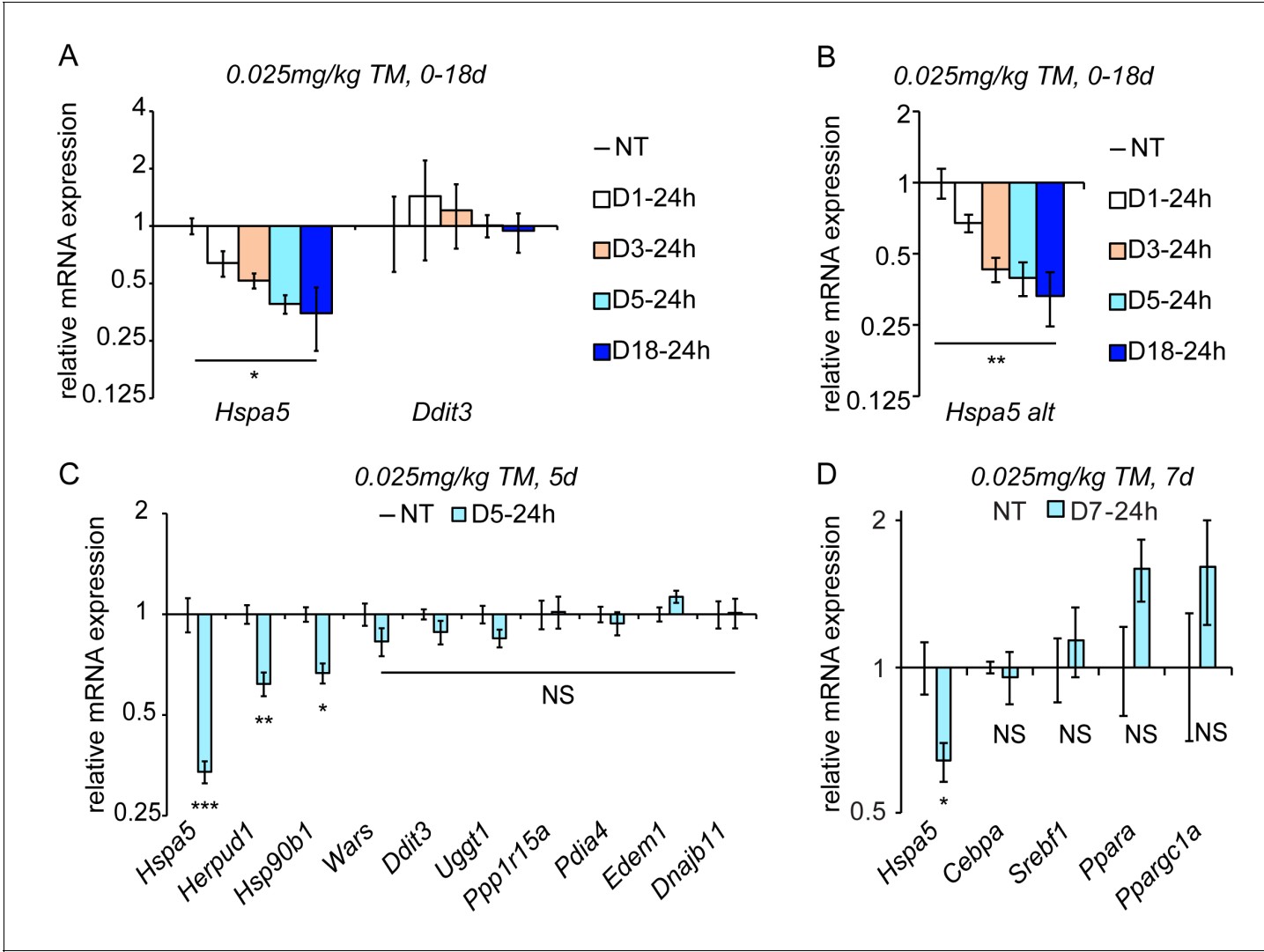

**Figure 2.** Chronic stress suppresses a subset of UPR target genes. (**A**) Mice were injected with 0.025 mg/kg TM daily for up to 18 days, and livers were collected 24 hr after the last injection as indicated. Expression of *Hspa5* and *Ddit3* was measured by qRT-PCR. Expression is given relative to vehicle-treated control mice. Data for all qRT-PCR experiments is shown on a log scale. n = 2–3 animals/group (**B**) Expression of *Hspa5* using an alternate primer pair spanning a different exon/intron junction confirms results from (**A**). (**C**) qRT-PCR of a group of UPR target genes from animals treated with vehicle or TM for 5d. n = 8 animals/group from two experiments. (**D**) Expression of the indicated metabolic genes was determined by qRT-PCR from an experiment similar to (**C**), except injections were performed for 7d. n = 4 animals/group.
The following source data is available for figure 2:

**Source data 1.** Contains raw and transformed Ct values for qRT-PCR experiments in *Figure 2A–D*.

sufficiently mild. Thus, we used these conditions to investigate how the UPR is regulated during such adaptation.

## Chronic stress suppresses select UPR targets

As the UPR is fundamentally a gene regulatory program, we examined the effects of chronic stress on UPR-dependent transcription. Despite *Ddit3* mRNA being upregulated by acute stress (*Figure 1A*), its expression returned to basal levels by 24 hr (D1-24h) and remained at similar levels at every subsequent time point, even after 18d of repeated treatment (*Figure 2A*). This finding mirrored the behavior of CHOP protein (*Figure 1F*). Surprisingly, however, *Hspa5* mRNA levels—while elevated by acute stress (*Figure 1A*)—were significantly diminished below basal levels by 24 hr and continued to diminish further through the first five days, remaining suppressed thereafter (*Figure 2A*). Similar results were obtained using qRT-PCR primers bridging a separate pair of exons in *Hspa5* mRNA, demonstrating that this suppression reflected total *Hspa5* mRNA levels rather than alternate splicing (*Figure 2B*).

Using the D5-24h point as representative of chronic stress conditions, we next surveyed other UPR-regulated gene targets. While the expression of most mRNAs, including *Ddit3*, was not significantly different in the chronic condition, a subset displayed a similar downregulation to *Hspa5*. These included mRNAs encoding the ER-associated degradation factor *Herpud1* and *Hsp90b1* (encoding the ER chaperone GRP94). (*Figure 2C*). We also assessed the expression of metabolic genes that we and others have previously shown to be suppressed by ER stress in the liver (*Rutkowski et al., 2008*; *Yamamoto et al., 2010*; *Arensdorf et al., 2013a*). While each of these was downregulated in the acute condition as expected (not shown), none was significantly suppressed in the chronic condition (*Figure 2D*). This finding provides evidence that the suppression of *Hspa5* and other mRNAs during chronic ER stress is mechanistically distinct from the pathways that suppress metabolic gene expression during acute stress.

## Chronic stress is associated with suppression of UPR-dependent transcriptional pathways

The observation that all of the assessed UPR targets either were unregulated or were suppressed in the chronic condition suggested that, despite persistent upregulation of at least some proteins such as BiP, transcriptional regulation was a more dynamic process and might be suppressed during chronic stress, at least by the 24 hr that elapsed between each chronic stress time point and its most proximate dose of TM. One of the commonalities among the genes that were suppressed below basal levels (*Hspa5*, *Herpud1*, *Hsp90b1*) is that each has been identified as a target of the ATF6α pathway of the UPR (*Yoshida et al., 2001*; *Yamamoto et al., 2004*; *Wu et al., 2007*). Thus, we hypothesized that UPR-dependent transcription must not only be deactivated to basal (i.e., unstressed) levels in the chronic condition, but also that at least ATF6α activity would be even further suppressed.

To test this hypothesis, we first assessed expression of the key UPR transcriptional regulators by immunoblot from liver nuclei. As a positive control, acute stress (8 hr treatment with 0.025 mg/kg TM as in *Figure 1A*) led to elevated expression of ATF4, XBP1$_{spl}$ and ATF6α$_{cl}$; however, all three were undetectable in the chronic condition (*Figure 3A*). This result was mirrored in a chromatin-immunoprecipitation (ChIP) experiment. For this experiment, the *Hspa5* promoter was recovered by an ATF6α antibody following 8 hr of treatment served as expected (*Yoshida et al., 1998*). In contrast, no ATF6α binding above background was detected in the chronic condition (*Figure 3B*). These ChIP data suggested that transcription of *Hspa5* was completely lost in the chronic condition, which was confirmed by ChIP directed against RNA Polymerase 2 (Pol2) at the *Hspa5* locus: As expected, Pol2 was recovered near the *Hspa5* transcriptional start site (−215 to +0), and this binding was unaffected by the presence or absence of ER stress (*Figure 3C*). This binding reflects Pol2 that is poised at the *Hspa5* promoter waiting to be engaged during the transcriptional activation process (*Muse et al., 2007*). In contrast, while Pol2 was abundantly recovered from the body of the *Hspa5* locus during acute stress (Intron 7, or +2727 to +2906, which reflects elongating polymerase), this binding was completely lost in the chronic condition (*Figure 3C*).

Taken together, these results demonstrate that ATF6α-dependent transcription (and possibly XBP1- and ATF4-dependent transcription as well) is silenced in the chronic condition. However, for

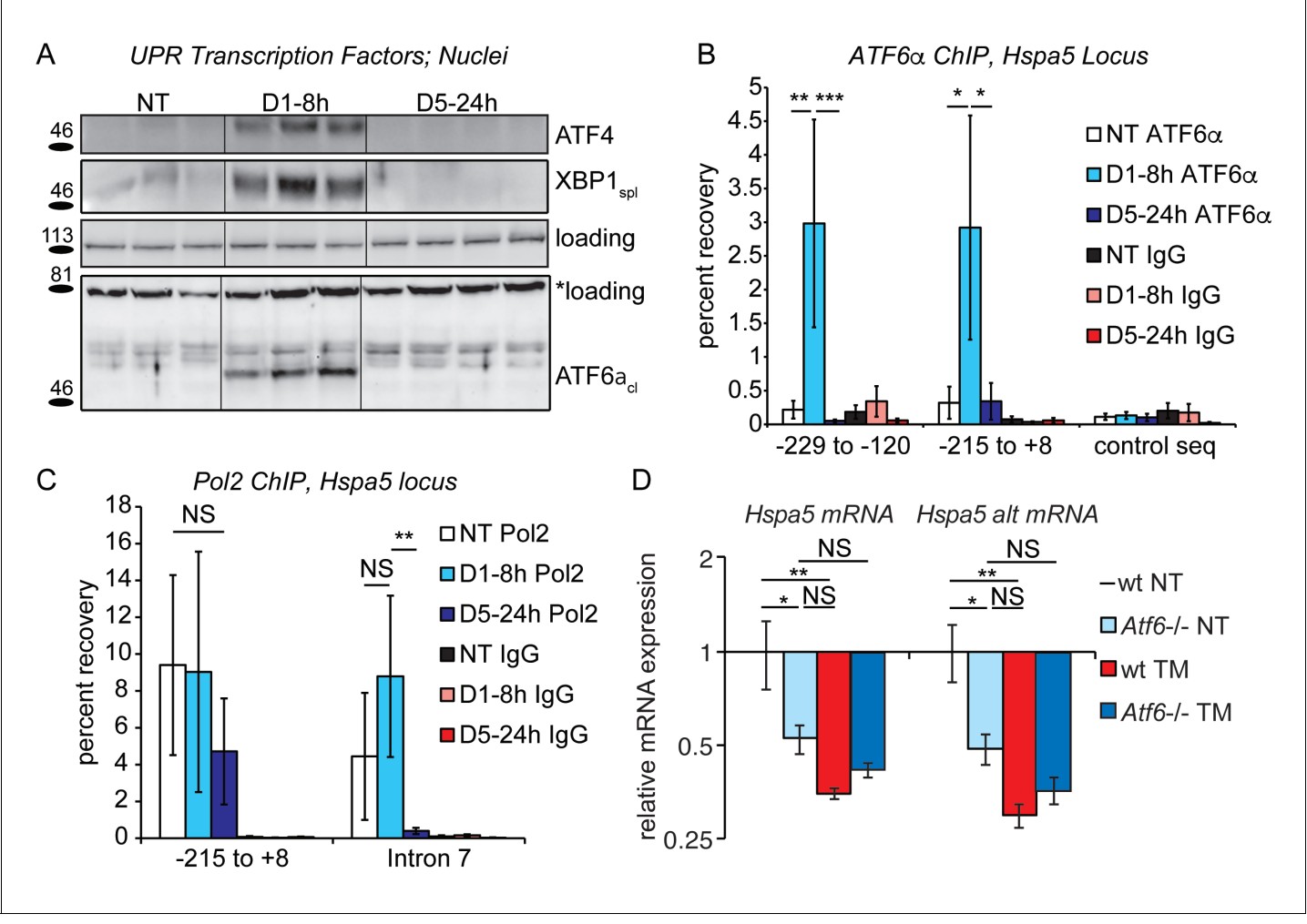

**Figure 3.** Chronic stress silences ATF6α-dependent transcription. (**A**) After treatment of mice with 0.025 mg/kg TM for either 8 hr or 5d, nuclei were isolated from liver lysates and probed with antibodies against ATF4, XBP1, or ATF6α. Specificity of all antibodies was confirmed by immunoblot from knockout or overexpression cells or liver lysates (not shown). The loading control for ATF4 and XBP1 blots was PARP. The loading control for the ATF6α immunblot is a nonspecific background band (*). (**B**) After TM treatment as in (**A**), control Ig or antibodies against ATF6α were used to immunopurify the *Hspa5* promoter region, which was then amplified by qPCR using primers directed against two overlapping regions in the promoter containing the ER stress elements (ERSEs) or a control sequence. Number is given relative to the transcriptional start site. n = 3–4 animals/group (**C**) Same as (**B**), except ChIP was performed using antibodies against RNA Polymerase 2. The −215 to + 8 regions detects poised polymerase and so Pol2 binding does not change, while the Intron seven region (+2727 to+2906) detects elongating Pol2. (**D**) Wild-type or *Atf6-/-* mice were treated with 0.025 mg/kg TM or vehicle for 5d. qRT-PCR expression of *Hspa5* in the liver was assessed using both primer sets. *Hspa5* expression was suppressed to comparable levels by either chronic stress or deletion of ATF6α, but chronic stress had no effect in *Atf6-/-* animals. n = 3–5 animals/group.

The following source data is available for figure 3:

**Source data 1.** Contains raw and transformed Ct values for qPCR and qRT-PCR experiments in *Figure 3B–D*.

this inhibition of transcription to result in suppression of *Hspa5* mRNA expression compared to untreated animals, ATF6α must contribute not only to stress-dependent transcription of *Hspa5* but also to the basal regulation of *Hspa5*. Confirming this prediction, we found that the basal expression of *Hspa5* in animals lacking ATF6α was reduced by approximately 50 percent relative to wild-type animals, and that chronic stress then had no further effect on *Hspa5* expression (*Figure 3D*). Thus, loss of ATF6α alone was sufficient to mimic chronic stress treatment. From these data, we conclude that at least the ATF6α pathway of the UPR is engaged in the liver absent exogenous stress, and that even this basal component can be inhibited upon chronic stress.

## Chronic stress is marked by cycles of activation and deactivation that suppress UPR-dependent transcription but possibly maintain RIDD

To this point, our data establish that under conditions of chronic stress, UPR-dependent transcription is attenuated, including, apparently, the contribution of ATF6α to basal *Hspa5* expression. This attenuation could come about in either of two ways. One possibility is that some fundamental feature of chronic stress renders the UPR increasingly refractory. This means that after 5 days of activation by successive bouts of stress the UPR would no longer be responsive; this is a 'preconditioning' model. The other possibility is an 'augmented deactivation' model in which the UPR is activated in full or nearly so upon each exposure to stress, but the temporal dynamics of its activation and resolution become progressively altered. In other words, the question is whether chronic stress mutes the activation of the UPR (*preconditioning*) or enhances its deactivation (*augmented deactivation*).

To distinguish between these possibilities, we examined the temporal response of the UPR in animals exposed to either the first dose of TM or the fifth dose (*Figure 4A*). Rather than following the production of UPR-regulated transcription factors as in *Figure 3*, we instead monitored activation of the UPR sensor IRE1α, which serves as a direct, robust, and sensitive readout for the capacity of the UPR to be activated. IRE1α was monitored using *Xbp1* mRNA splicing (*Figure 4B*). This experiment demonstrated that the UPR remained essentially as stress-responsive on the fifth day as on the first; splicing of *Xbp1* 8 hr after the first injection and 8 hr after the fifth were induced to comparable levels. Supporting the idea that the UPR remained responsive, repeated TM treatment continued to induce short-term lipid dysregulation, as evidenced by abundant cytoplasmic vacuoles in D5-8h livers (*Figure 4C*). Yet the *deactivation* of the UPR, or at least the *Xbp1* splicing activity of IRE1α, proceeded more rapidly on day five than on day 1; substantial *Xbp1* splicing was seen 14 hr after the first injection but not 14 hr after the fifth (*Figure 4B*). Likewise, as shown in *Figure 1D*, lipid dysregulation was observed 24 hr after the first injection but not 24 hr after the fifth. We also monitored *Hspa5* mRNA expression over a similar time course, and found that its expression mirrored this pattern—namely, that *Hspa5* expression was well-induced after both treatments, but that its attenuation proceeded more rapidly in the chronic condition (*Figure 4D*). Similar results were also obtained for *Hsp90b1* (encoding GRP94) mRNA (not shown). Collectively, our data support the *augmented deactivation* model described above. Thus, the chronic treatment opens a window into the mechanisms of UPR deactivation, about which relatively little is known since ER stress is usually studied in the context of stresses sufficiently severe as to kill cells.

These experiments also yielded an additional insight, which was that the rapid loss of *Hspa5* mRNA by the fifth day might not be accounted for solely by the loss of *Hspa5* transcription. *Hspa5* mRNA levels decreased more than 30-fold between 8 hr and 24 hr on the fifth day, for a half-life of approximately 3 hr (*Figure 4D*). However, by monitoring *Hspa5* expression after treatment of primary hepatocytes with the transcription inhibitor Actinomycin D, we measured *Hspa5* mRNA half-life as 8.2 hr after treatment with TM (*Figure 4E*), consistent with our previous measurement in MEFs (*Rutkowski et al., 2006*). The simplest explanation for these results is that deactivation of the UPR is accompanied by stimulated degradation of *Hspa5* mRNA, for even the loss of *Hspa5* that occurs on the first day of treatment (*Figure 4D*) is too rapid to be accounted for by inhibition of transcription alone.

A candidate pathway for stimulated degradation of *Hspa5* mRNA is Regulated IRE1-Dependent Decay (RIDD), which is the process by which activated IRE1 directly degrades ER-associated mRNAs (*Hollien et al., 2009*). This process is thought to be counter-adaptive, facilitating cell death during severe ER stress (*Han et al., 2009*; *Upton et al., 2012*). However, it has also been implicated in metabolic regulation in the liver (*Cretenet et al., 2010*; *So et al., 2012*), implying that perhaps its function is more nuanced. *Hspa5*, *Hsp90b1*, and *Herpud1* mRNAs would be expected to localize to the ER by virtue of encoding proteins bearing ER targeting signals. *Hspa5* at least has also been proposed to be a RIDD target, though with unclear functional significance (*Han et al., 2009*).

To test whether RIDD was active in the chronic stress condition, we assessed the expression of the well-validated RIDD target *Bloc1s1* (aka *Blos1*). Among putative RIDD targets, only *Bloc1s1* has been documented in multiple reports, and in highly diverse cell types (*Bright et al., 2015*). We confirmed that *Bloc1s1* is a *bona fide* RIDD target in MEFs lacking IRE1α, in which its downregulation by ER stress was completely lost (*Figure 4F*). As expected, *Bloc1s1* expression was reduced in the liver by acute (8 hr) TM treatment (*Figure 4G*). Furthermore, while *Bloc1s1* expression returned to normal

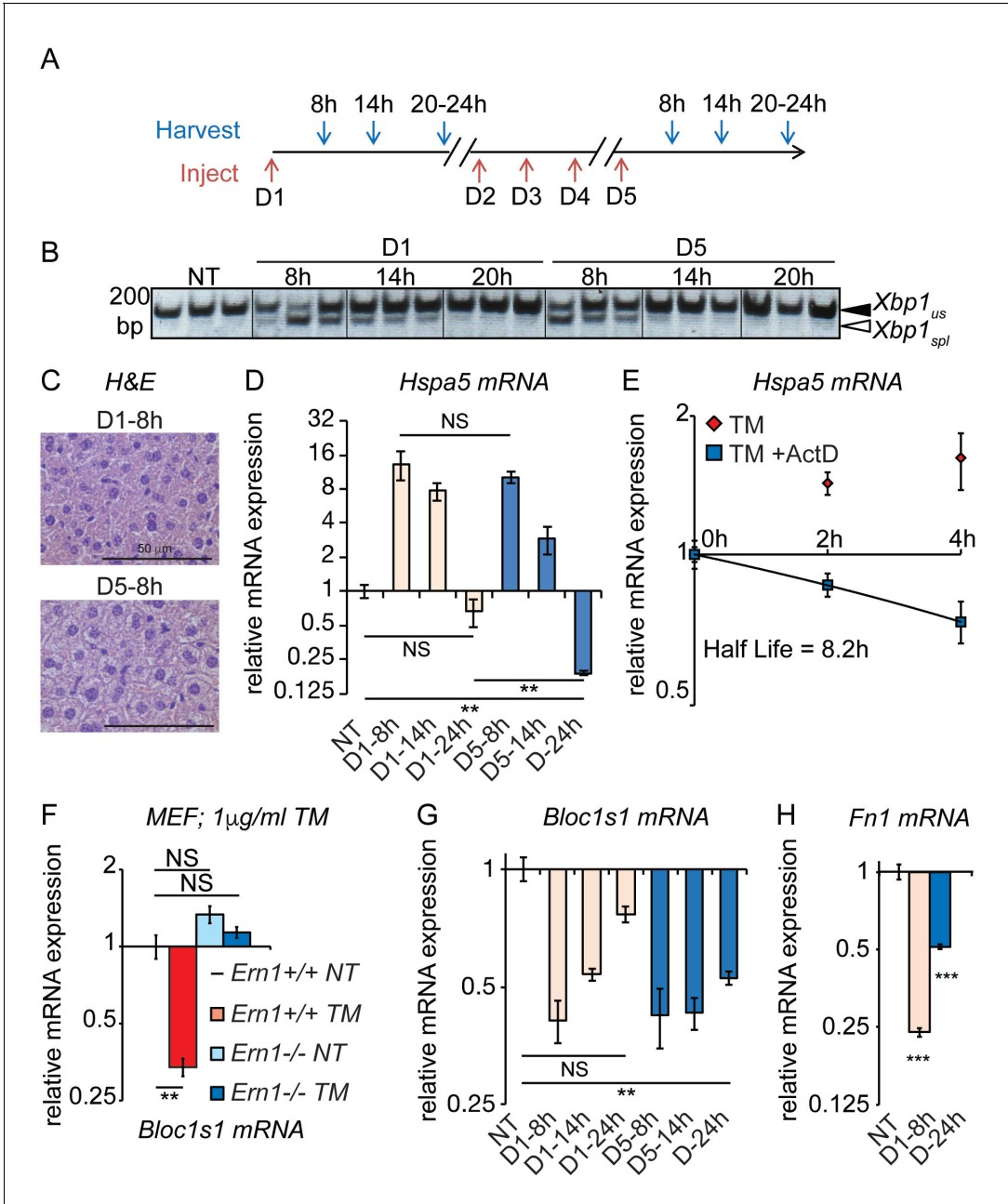

**Figure 4.** Accelerated degradation diminishes *Hspa5* mRNA levels during chronic stress. (A) Schematic showing treatment protocol; livers were harvested 8, 14, or 20–24 hr after either the first TM injection or the fifth. (B) Conventional RT-PCR was used to distinguished spliced (spl) from unspliced (us) *Xbp1* mRNA in samples treated as in (A). Image is inverted black-to-white for greater visual clarity. Each lane represents a separate animal. (C) H and E staining of liver sections harvested at the D1-8h or D5-8h timepoints. (D) *Hspa5* mRNA expression was assessed by qRT-PCR from samples treated as in (B). n = 3–4 animals per group (E) Primary hepatocytes were isolated from a wild-type mouse, and treated with TM in the presence or absence of actinomycin D (ActD) to inhibit transcription as described in Materials and methods. *Hspa5* half-life was calculated from these data. n = 3 plates/group (F) Wild-type or *Ern1-/-* (lacking IRE1α) mouse embryonic fibroblasts (MEFs) were treated with TM for 8 hr and expression of the RIDD target *Bloc1s1* was determined by qRT-PCR. n = 3 plates/group (G) Animals were treated as in (A) and *Bloc1s1* expression was detected by qRT-PCR. (H) Animals were treated for 8 hr or 5d with 0.025 mg/kg TM. Expression of *Fn1* was determined by qRT-PCR. n = 4 animals/group.

The following source data is available for figure 4:

**Source data 1.** Contains raw and transformed Ct values for qRT-PCR experiments in *Figure 4D–H*.

levels 24 hr after the first TM treatment, it remained suppressed 24 hr after the fifth treatment (*Figure 4G*). Suppression of a RIDD target in the chronic condition was also seen for *Fn1*, which we found to be a potential RIDD target in the liver based on microarray data from wild-type and liver-specific *Ern1-/-* (encoding IRE1α) mice treated with TM (*Zhang et al., 2011*) (*Figure 4H*).

The behavior of *Bloc1s1* and *Fn1* is in contrast to genes such as *Ddit3*, *Wars*, etc., that were transcriptionally induced by acute ER stress, but which did not remain induced in the chronic condition (*Figure 2C*), as well as to metabolic genes that are transcriptionally repressed by acute ER stress, which did not remain repressed in the chronic condition (*Figure 2D*). Computational modeling suggests that mRNAs whose expression is regulated by stimulated degradation should return to basal expression levels much more rapidly than those whose regulation is transcriptional, when both mechanisms are deactivated (*Arensdorf et al., 2013b*). Thus, this persistence of *Bloc1s1* and *Fn1* suppression might suggest that, in contrast to the pathways of transcriptional regulation, RIDD activity persists through the cycles of UPR activation and deactivation that characterize the chronic stress state, even while the *Xbp1* splicing activity of IRE1α is attenuated. Conclusively demonstrating that RIDD is active under these conditions will ultimately require replacement of endogenous IRE1α by a form in which the *Xbp1* splicing and RIDD activities can be separated and modulated (*Han et al., 2009*). Thus, while the hypothesis that RIDD activity reduces the expression of *Hspa5* and other genes is an appealing possible mechanism, other degradative pathways are possible.

## Overexpression of BiP is sufficient to suppress Hspa5 (BiP) and Hsp90b1 mRNA

Our results to this point suggest that with each successive exposure to stress, the UPR becomes more efficient at deactivation, even to the point of overshooting, while still retaining its activation capacity. The mechanisms of deactivation, however, are poorly understood. One pathway by which the UPR is deactivated is upregulation of BiP, which turns off the response in two ways: by improving ER protein folding (in concert with the many other factors regulated by the UPR); and by directly binding to the UPR stress sensors (IRE1α, ATF6α, and PERK) and repressing their activation (*Bertolotti et al., 2000*; *Shen et al., 2002*).

In the chronic condition, *Hspa5* mRNA was suppressed despite ongoing upregulation of BiP protein (*Figure 1F*); the same was true for GRP94 (not shown). Given the repeated cycles of activation that characterize the chronic response (*Figure 4*), it is perhaps unsurprising that BiP accumulates to the extent that it does; we have previously measured the half-life of BiP protein in MEFs at approximately two days (*Rutkowski et al., 2006*), and it appears to also be long-lived in hepatocytes (not shown). This persistent BiP protein expression thus provides a potential means by which UPR deactivation could be accelerated.

To test this hypothesis, we ectopically overexpressed BiP protein in the liver using recombinant adenovirus (ad-BiP, aka AdGRP78) (*Young et al., 2012*). After injection of ad-BiP, we observed a significant increase in BiP protein expression compared to ad-GFP expressing animals (*Figure 5A*). qRT-PCR confirmed the efficacy of exogenous BiP overexpression (*Figure 5B*). Using primers that detect the 5' UTR of *Hspa5* mRNA (and thus amplify only endogenous *Hspa5* and not exogenous), we found that, like chronic stress, BiP protein overexpression suppressed *Hspa5* mRNA expression (*Figure 5C*). Similar results were obtained for *Hsp90b1*, while overexpression had no effect on *Ddit3* expression (*Figure 5C*). Surprisingly, *Bloc1s1* expression was also suppressed under these conditions, again mirroring the chronic condition (*Figure 5D*). This finding hints that the apparent persistence of RIDD in the chronic condition (*Figure 4G,H*) might reflect a unique sustainment of that activity of IRE1α by BiP during recovery from stress when the transcriptional limbs of the UPR are suppressed.

The effects of BiP overexpression on *Hspa5* and *Hsp90b1* mRNA could be due either to enhanced protein folding in the ER or to direct effects of BiP on the UPR sensors independent of the protein folding capacity. We found that treating mice with the pharmacological chaperone TUDCA (*Ozcan et al., 2006*)—which reduced upregulation of ER stress markers by TM approximately 2-fold—had no effect on its own on any UPR gene tested (*Figure 5E*). These data show that improved ER protein folding capacity was not sufficient to suppress *Hspa5* expression. While they do not completely rule out the possibility that improved ER protein folding ultimately drives *Hspa5* suppression in the chronic condition, they nonetheless lead us to favor a model whereby BiP directly modulates the activities of at least ATF6α and IRE1α.

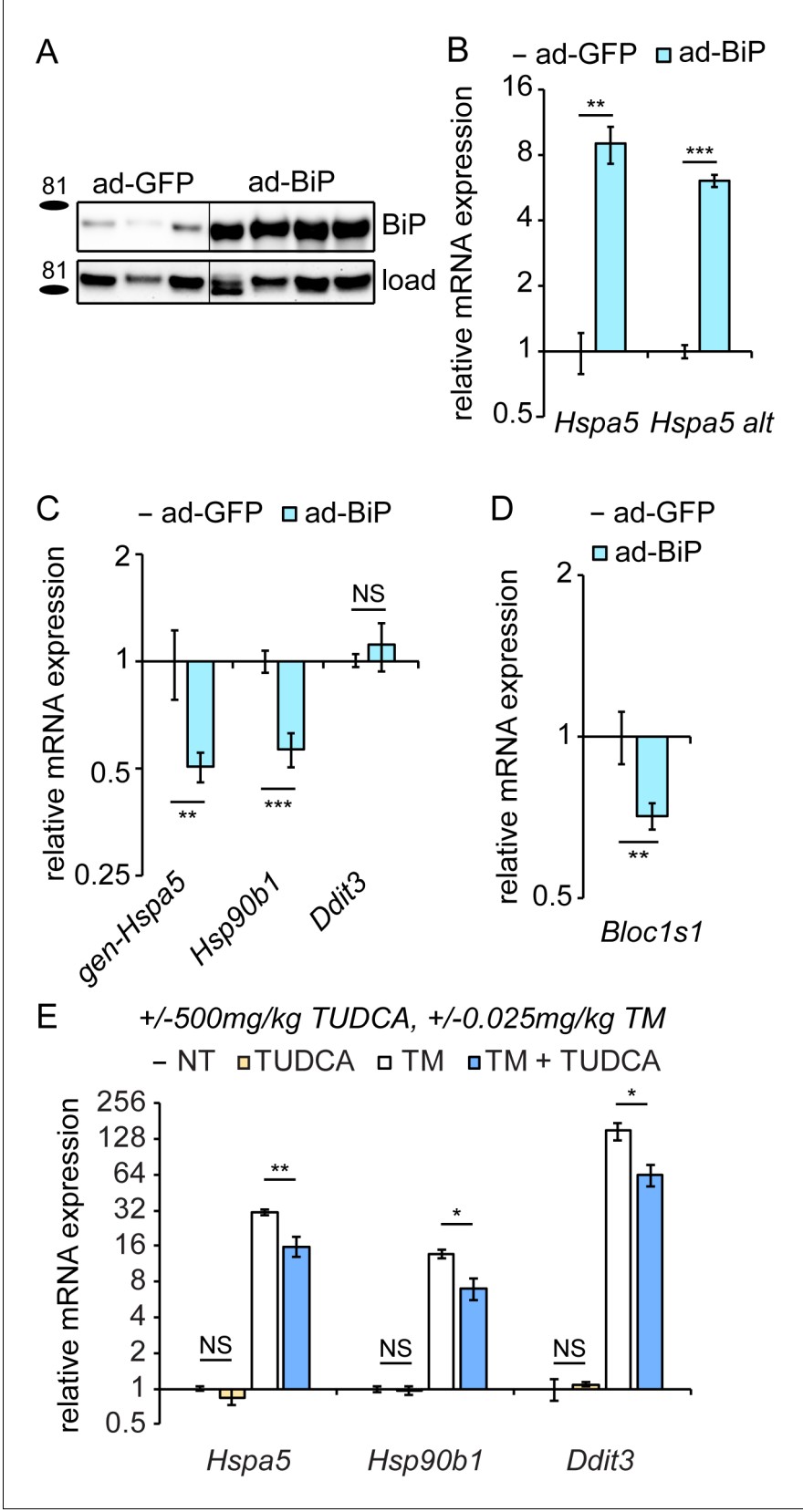

**Figure 5.** Overexpression of exogenous BiP is sufficient to repress endogenous *Hspa5* expression. (**A**) Animals were injected with recombinant adenovirus expressing either GFP or BiP. 5d after injection, mice were sacrificed

*Figure 5 continued on next page*

*Figure 5 continued*
and BiP was probed by immunoblot. Loading control was calnexin. (B) Both primer sets detected elevated *Hspa5* expression in ad-BiP mice, which is attributable to the contribution of the exogenous *Hspa5*. n = 8 animals/group for regular *Hspa5* primers and 3–4 animals/group for *Hspa5* alternate primers. (C, D) Expression of endogenous *Hspa5* from the genomic locus (gen-*Hspa5*), *Hsp90b1*, or *Ddit3* (C) or *Bloc1s1* (D) was assessed in ad-GFP and ad-BiP animals by qRT-PCR. n = 8 animals/group from two experiments. (E) Wild-type animals were treated for 10d with 500 mg/kg TUDCA, and then for 8 hr with 0.025 mg/kg TM, and expression of the indicated genes was detected by qRT-PCR. TUDCA was sufficient to reduce stress-induced expression of these genes approximately two-fold or more, but not to suppress basal *Hspa5* expression. n = 3–6 animals/group.
The following source data is available for figure 5:

**Source data 1.** Contains raw and transformed Ct values for qRT-PCR experiments in *Figure 5B–E*.

## Suppression of Hspa5 expression and activation of RIDD are mirrored in the livers of genetically obese animals

To what extent are the regulatory events described here relevant to physiological chronic stresses? Repeated dosing with TM, while effective as a tool for inducing ER stress, is nonetheless doubtless much more robust and focal a stress than anything commonly encountered physiologically. However, the cyclic nature of the stimulus—bouts of stress followed by periods of recovery—might to some extent mirror the stress caused by metabolic flux, which is tied to feeding and fasting cycles and is thus by its nature cyclic. The notion that overnutrition—i.e., consumption of too much food per meal and/or too many meals—might represent a chronic ER stress was first suggested in 2004, in seminal work demonstrating that ER stress was observed in the livers of obese mice, and was tied to hepatic insulin resistance (*Ozcan et al., 2004*). Thus, we considered the possibility that feeding is capable of eliciting the same changes in gene regulation as seen in the TM-induced chronic condition.

Consistent with previous reports (*Oyadomari et al., 2008*; *Pfaffenbach et al., 2010*), we found that feeding elicited ER stress in mouse livers (*Figure 6A*). Interestingly, at least at this timepoint the effect was much more robust for some targets (*Hspa5*, *Hsp90b1*, and *Xbp1*) than others; in fact, targets of the PERK pathway (*Ddit3*, *Ppp1r15a* [encoding GADD34], and *Wars*) were not upregulated to a statistically significant extent under these conditions. Feeding also elicited splicing of *Xbp1* mRNA as measured by qRT-PCR using primers that preferentially amplify only the spliced form (*Xbp1(s)*, *Figure 6A*), albeit to a very modest extent (once the change in total *Xbp1* levels is accounted for). While it has been proposed that a high fat diet is intrinsically stress-inducing (*Ozcan et al., 2004*), at least in this instance we found no difference between the level of stress induced by diets containing either high fat or no fat (*Figure 6A*). This result indicates that nutrient intake itself elicits ER stress, and is consistent with the idea that the stress is driven by increased anabolism (*Ozcan et al., 2008*) rather than fat per se. As we speculated, this stress is modest compared to that induced by TM, for which most UPR targets were induced to an extent that was at least an order of magnitude greater than feeding (*Figure 6B*).

If feeding represents an ER stress, then repeated feeding should represent a chronic stress, particularly if it outpaces the ability of the organelle to compensate. Thus, we examined gene expression in $Lep^{ob/ob}$ or $Lepr^{db/db}$ mice, which are obese due to mutations in leptin or the leptin receptor, respectively, that compromise appetite control and lead to overfeeding (*Campfield et al., 1995*; *Halaas et al., 1995*; *Pelleymounter et al., 1995*; *Chen et al., 1996*; *Lee et al., 1996*). We observed a similar pattern of gene regulation in these animals as in animals treated with chronic TM; *Hspa5* and *Hsp90b1* were both significantly suppressed (*Figure 6C*), while other UPR targets were not suppressed (and, indeed, appeared to be upregulated; *Figure 6D*). In addition, again mirroring the results seen in TM-treated animals, expression of *Bloc1s1* was reduced in $Lepr^{db/db}$ mice and potentially in $Lep^{ob/ob}$ mice as well, and *Fn1* was strongly suppressed in both (*Figure 6E*). These results suggest that suppression of the mRNA expression of *Hspa5* and *Hsp90b1* characterizes both pharmacological and dietary models of chronic stress. Though they do not yet establish that suppression proceeds by the same mechanism in both, they highlight the potential for the experimental induction of chronic ER stress to be a useful tool for understanding how gene regulatory patterns are altered by chronic physiological stresses.

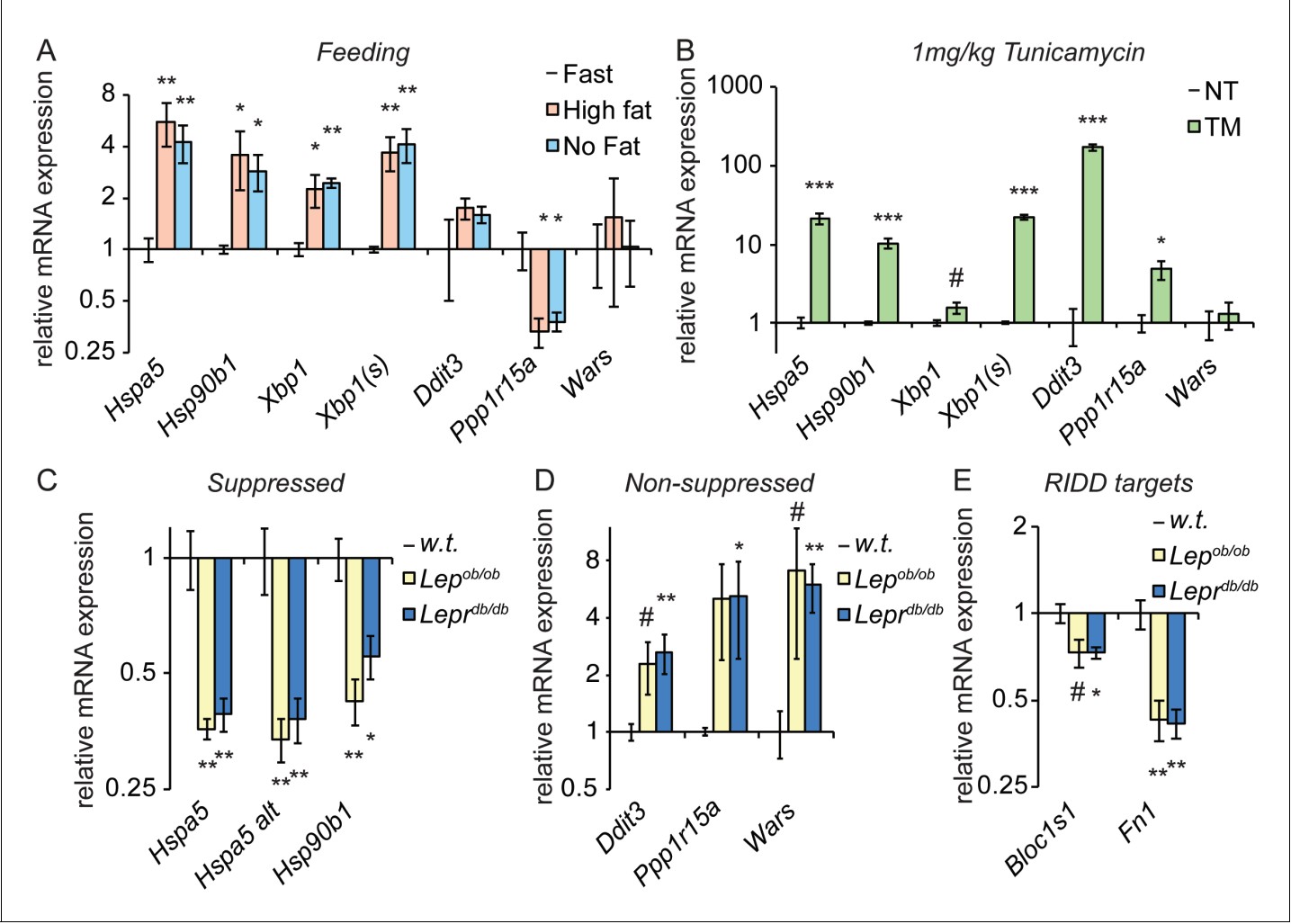

**Figure 6.** Genetically-induced obesity phenocopies chronic stress. (A) Wild-type animals were fasted overnight, and then provided food containing either 45% fat or no fat for 4 hr. Expression of the indicated genes was assessed by qRT-PCR. n = 3 animals/group (B) Wild-type animals were treated with 1 mg/kg TM for 4 hr, and the same genes as in (A) were detected by qRT-PCR. #; p<0.1. n = 3 animals/group (C–E) Livers from five month-old female $Lep^{ob/ob}$ or $Lepr^{db/db}$ mice or age-matched wild-type mice were probed for expression of genes that were suppressed by chronic stress (C) or not-suppressed (D), or of the RIDD target $Bloc1s1$ and the putative RIDD target $Fn1$ (E). n = 4 animals/group.

The following source data is available for figure 6:

**Source data 1.** Contains raw and transformed Ct values for qRT-PCR experiments in *Figure 6A–E*.

## Discussion

In this paper, we have described the first, to our knowledge, experimental reconstitution of chronic, tractable ER stress in a mammal in vivo. This work builds upon previous efforts to recreate chronic ER stress in fish larvae (*Cinaroglu et al., 2011*), and to elicit chronic ER stress in mammals by overexpression of misfolded ER client proteins or deletion of ER chaperones (e.g., [*Ji et al., 2011*; *Zode et al., 2011*; *Kawasaki et al., 2015*]). We have used this system as a tool to probe how UPR signaling becomes altered during persistent and/or repeated activation. Our investigation has several novel implications for the mechanisms of UPR signaling.

The data presented here lead us to propose a working model to account for the altered expression of *Hspa5 (Bip)* and other UPR transcriptional targets during chronic stress (*Figure 7*). In the basal (i.e., nominally 'unstressed') state, activation of ATF6α contributes to the expression of *Hspa5* mRNA. We arrive at this conclusion because expression of *Hspa5* was diminished in the absence of

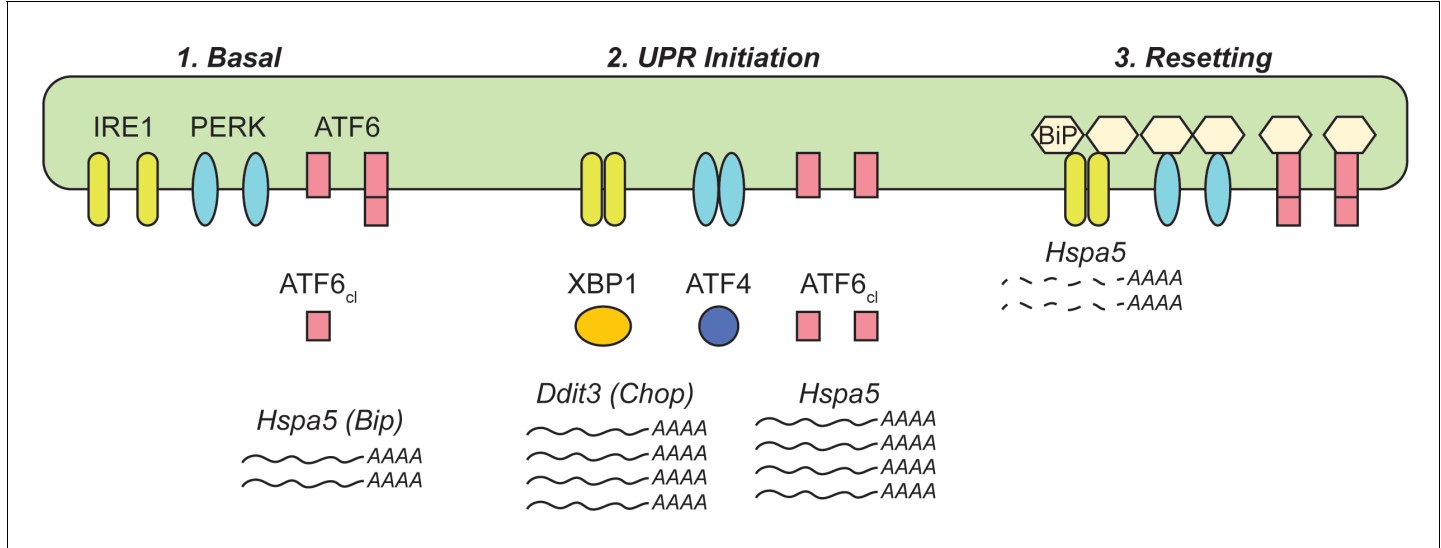

**Figure 7.** Model for UPR dynamics during chronic stress. See Discussion for details.

ATF6α even without an exogenous stress (*Figure 3D*). This finding implies that normal physiological function in the liver entails periods of UPR activation—perhaps not surprisingly, given that feeding itself activates the UPR (*Figure 6A*). BiP is one of the most abundant proteins in the ER lumen (*Gething, 1999*), and other non-stress-dependent mechanisms contribute to its basal regulation as well (*Resendez et al., 1988*). Upon exposure to a conventional ER stress, all three limbs of the UPR are activated (*Rutkowski et al., 2006*), leading to robust production of stress-specific target transcripts such as *Ddit3 (Chop)*, as well as augmentation of transcripts encoding ER chaperones, such as *Hspa5* and *Hsp90b1 (Grp94)*. The attendant improvement of ER protein folding then presents a problem for the cell: how to avoid overproduction of UPR targets. As the UPR stress-sensing molecules become deactivated, naturally labile factors such as CHOP (both mRNA and protein) are readily lost (*Rutkowski et al., 2006*). However, ER chaperones such as BiP and GRP94 are long-lived at both the protein and mRNA levels (*Rutkowski et al., 2006*), and continued overproduction of these proteins would presumably greatly tax cellular resources and potentially constitute their own burden on the organelle. Thus, BiP overexpression appears to initiate a negative feedback loop, both suppressing UPR activity (including the basal activity of ATF6α) and also indirectly stimulating the degradation of its own mRNA. In the context of incidental exposure to stress, or of the low-level stresses that occur during normal physiological function, we presume that this depression of *Hspa5* mRNA levels is transient. However, during repeated stress, we propose that it constitutes a resetting to a new quasi-stable setpoint for UPR transcriptional regulation.

One of the founding observations of the UPR field was that activation of the ER stress sensing molecules required their dissociation from BiP in both yeast and mammals (*Bertolotti et al., 2000*; *Shen et al., 2002*). Originally, the purpose attributed to BiP binding was to hold the stress sensors in a quiescent state and so to play a role in UPR activation. However, the suggestion that at least yeast Ire1p can be activated by direct binding to unfolded proteins (*Kimata et al., 2004*; *Gardner and Walter, 2011*; *Gardner et al., 2013*) has cast doubt on this model. Our data raise the possibility that perhaps the role of BiP binding is to instead modulate UPR *deactivation*.

The objective of this work from the outset was to develop an experimental system for eliciting chronic ER stress specifically, in order to be able to determine how the regulation and output of the response changed under such conditions and so identify fingerprints of chronic ER stress associated with disease states. An important caveat is that most agents other than TM that specifically induce ER stress in vitro fail to do so easily or effectively in vivo. While relatively specific in inducing ER stress due to its blockage of N-linked glycosylation, TM also is capable of disrupting carbohydrate metabolism (*Olden et al., 1979*) and protein palmitoylation (*Patterson and Skene, 1995*), although whether it does so at the doses used here is unclear. In addition, it is conceivable that the chronic

stress phenotype is caused not by the ER stress-inducing effect of TM, but by underglycosylation of some specific key protein. However, we do not favor this alternative explanation in part because the degree of protein underglycosylation under these conditions is relatively modest (e.g., *Figure 1E*), and also because of the ability of BiP overexpression to phenocopy *Hspa5* downregulation (*Figure 5*).

In addition, the strikingly similar phenotype in *Lep*^ob/ob and *Lepr*^db/db mice vouches for this approach. One difference between the chronic condition on one hand and the steady-state in obese mice on the other is that the latter do not appear to express elevated BiP protein levels (not shown), which at face value seems at odds with the idea that elevated BiP protein is needed to both suppress ATF6α and activate or perpetuate RIDD. However, as we have demonstrated, the magnitude of ER stress elicited by feeding is considerably lower than that elicited by tunicamycin (*Figure 6*; also compare *Figure 6A* with *Figure 4D*), yet also much more repetitive than the once-daily TM challenge. It is plausible that the less robust but more constant stimulus of overfeeding results in modest alterations in expression of BiP protein that have little effect on *Hspa5* mRNA over the course of meals or days but great effect over the course of months—essentially that small increases in BiP protein levels could produce small decreases in BiP mRNA expression that accumulate over time. It is possible that further moderating the dose of TM and lengthening its extent (from days to weeks/months) would more closely mimic the phenotype observed in obese animals, although this would be experimentally cumbersome. It is also worth noting that the suppression of *Hspa5* and *Hsp90b1* mRNAs might be achieved by other mechanisms that suppress ATF6α activation while perpetuating RIDD. For instance, it has been reported that ATF6α expression is lost in obese animals (*Wang et al., 2009*). Along these lines, we observed that *Atf6* mRNA is also significantly downregulated by chronic stress, albeit not as robustly as *Hspa5* (not shown), raising the possibility of a positive feedback loop for suppressing *Hspa5* expression even as BiP protein levels subside. Alternatively, posttranslational modifications that alter BiP's interactions with its binding partners (*Chambers et al., 2012*; *Wang et al., 2014*; *Preissler et al., 2015*) might also play a role. Ultimately, the clearest way to test whether the mechanism we describe here is at work in obese animals might be to intercross these animals with *Atf6*-/- animals; as when given repeated TM injections, these animals should have basally suppressed *Hspa5* mRNA expression even at an early age, but then there should be no further suppression as the animals age and become obese. Even if the mechanism turns out to be entirely distinct, the description here of suppressed expression of *Hspa5*, and *Hsp90b1* and potentially elevated RIDD activity in these animals has potential implications for their ability to maintain ongoing ER homeostasis. It is conceivable that an altered UPR 'setpoint' with reduced expression of *Hspa5* and other UPR mRNA targets could render the animals more sensitive to ER stress-induced cell death and exacerbate the hepatic inflammation associated with obesity (*Gregor and Hotamisligil, 2011*).

A surprising conclusion from this work is that RIDD might remain active in the chronic condition, even as UPR-dependent transcriptional signaling is shut off. Admittedly, conclusively demonstrating RIDD activity in vivo is challenging; here we have relied on repression of the well-documented RIDD target *Bloc1s1* as the primary readout for RIDD activity. *Bloc1s1* expression was repressed in the chronic condition (*Figure 4G*), upon BiP overexpression (*Figure 5D*), and in *Lepr*^db/db animals (*Figure 6E*). Phos-tag immunoblotting showed substantial phosphorylation of IRE1α only in animals treated with a high acute dose of TM, and antibodies purported to be specific for phospho-IRE1 did not produce bands that were absent in cells lacking IRE1α in control experiments (not shown). These limitations made it impossible to directly test whether IRE1α remained phosphorylated in the chronic condition. In addition, there is as yet no functional assay for confirming that RIDD is active in vivo. Rather, substrates are generally identified as putative RIDD targets based on an in vitro assay using purified IRE1α (*Hollien and Weissman, 2006*; *Hollien et al., 2009*). Then, their RIDD dependence is confirmed by their suppression during ER stress in an IRE1α-dependent XBP1-independent manner. However, *Hspa5* and *Hsp90b1* are both also transcriptionally upregulated by XBP1 (*Lee et al., 2003*), which would likely confound analysis of their regulation in animals lacking IRE1α. In any case, our data hint at a previously unappreciated role for RIDD, which has generally been considered an anti-adaptive signaling mechanism to this point (*Han et al., 2009*; *Lerner et al., 2012*; *Upton et al., 2012*; *Maurel et al., 2014*). The purpose of the clearance of ER-resident proteins might not so much be to neuter the organelle and hasten its dysfunction during severe stress (*Maurel et al., 2014*) as to help turn the response off once stress has been overcome. If RIDD is indeed active under these

conditions, such a role could help explain the previously puzzling finding that *Hspa5* is a potential RIDD target (*Han et al., 2009*).

Assuming RIDD is active during chronic stress, our data provide further evidence that the *Xbp1* splicing and RIDD activities of IRE1α can be dissociated (*Lin et al., 2007*; *Han et al., 2008*). They raise the possibility that this dissociation occurs not necessarily to separate adaptive from apoptotic outputs so much as to separate activation from deactivation. Given that most stresses encountered by most cells most of the time are likely to be mild and transient—such as that encountered by feeding—the death-accelerating function of RIDD might simply be a byproduct of a normal physiological role in restoring the UPR rheostat. Indeed, a parallel can be drawn with the UPR-regulated transcription factor CHOP which, though it clearly promotes cell death during severe stress (*Zinszner et al., 1998*) also might promote normal cellular function during milder stress both by potentiating the dephosphorylation of eIF2α (*Marciniak et al., 2004*) and by regulating lipid metabolism (*Tyra et al., 2012*; *Chikka et al., 2013*) toward the ultimate improvement of ER function.

The approach we took to experimentally reconstituting chronic ER stress in vivo was conceptually analogous to our earlier reconstitution of chronic stress in vitro in MEFs (*Rutkowski et al., 2006*). However, neither MEFs nor even cultured hepatocytes showed the same suppression of *Hspa5* expression as seen in vivo. This discrepancy points to the existence of other as yet unappreciated factors that influence UPR output in the liver. Indeed, it is now becoming clear that, overlaid atop the structural framework of the canonical UPR lie modulatory pathways for altering UPR sensitivity and output, with potential implications for signaling during chronic stress. These include posttranslational modifications either of the stress sensors—such as S-nitrosylation of IRE1α (*Yang et al., 2015*)—or of ER chaperones, such as ADP ribosylation (*Chambers et al., 2012*) or AMPylation (*Preissler et al., 2015*) of BiP. It will be interesting to test whether any of these modifications is active during the chronic condition.

Finally, our results underscore the remarkable capacity of the UPR to adapt to chronic stress. While the severe experimental stresses that lead to massive cell death and organ dysfunction have proven useful in elucidating the signaling capabilities of the UPR when activated to its maximum, they yield little about the ebb and flow of the response at it is most likely to be activated in physiological scenarios. We hope that our approach will stimulate further efforts to more closely mimic physiological stresses in vivo, as these will be essential in linking chronic ER stress to human disease.

## Materials and methods

### Animal experiments

All protocols for animal use were reviewed and approved by the University Committee on Use and Care of Animals at the University of Iowa. Animals were fed standard rodent chow and housed in a controlled environment with 12 hr light and dark cycles. Animals used were of both genders unless otherwise noted, with control and experimental groups having similar composition. Animal numbers needed for each experiment were determined based on previous experiments. Animals were fasted for 4 hr prior to sacrifice, which was carried out during the lights-on period. *Atf6-/-* (RRID:MGI: 3723589) (*Wu et al., 2007*) animals have been backcrossed into the C57BL/6J strain (RRID:IMSR_ JAX:000664) for >10 generations. For chronic TM (EMD Millipore, Billerica, MA) treatment, 8 to 12 week old *C57BL/6J* or *Atf6-/-* mice were injected once per day (~1 hr after lights-on) intraperitoneally with vehicle or the indicated dose of Tunicamycin dissolved in 150 mM dextrose or PBS and were sacrificed at various timepoints after last injection. For TUDCA (EMD Millipore) treatment, animals were injected IP with vehicle or 500 mg/kg TUDCA in PBS once daily, 1 hr after lights-on for 10 days prior to sacrifice. 8 hr prior to sacrifice (3 hr after lights-on), animals were injected with TM or vehicle. For high fat/no fat feeding, animals were habituated for 5d to a standard defined diet (D12450B, Research Diets, New Brunswick, NJ), fasted 20 hr, and then either injected with 1 mg/kg TM for 4 hr or provided access to a high fat diet (D12451, 45% fat) or a no-fat diet (D10062804, 0% fat) for 4 hr. *Lep^{ob/ob}* and *Lepr^{db/db}* samples were taken from five month-old female mice.

Harvested liver tissue pieces were either frozen immediately for RNA or protein analysis, fixed in formalin (for histology), processed for nuclear isolation, or minced and fixed in formaldehyde (for ChIP). Nuclei were isolated as described (*Rutkowski et al., 2008*). For histological analysis, formalin-fixed tissues were embedded in paraffin, sectioned, and stained with hematoxylin and eosin.

Antibodies used for immunoblot were as in (*Chikka et al., 2013*), and additionally as follows: ATF4 (sc200, Santa Cruz, Dallas, TX; RRID:AB_2058752); XBP1 (sc7160, Santa Cruz; RRID:AB_794171); ATF6α (courtesy of R. Kaufman). Antibodies were generally incubated on blots in TBS-Tween + 5% milk at room temperature for 1 hr except for ATF6α antibody, which was incubated in PBS + milk (no Tween-20) at 4°C overnight.

## Cell experiments

MEFs were harvested as described previously (*Lee et al., 2002*) from *Ern1-/-* or *Ern1+/+* mouse embryos (RRID:MGI:3723591). Genotype was confirmed by PCR. Isolation of primary hepatocytes was as in (*Fan et al., 2004*) with minor modifications. Hepatocytes were isolated from *C57BL/6J* mice. Mice were anesthetized with isofluorane. The liver was perfused through the portal vein first with Perfusion Buffer Solution and then with Liver Digest Medium. Formulations were as follows: Liver Perfusion Media: HBSS, no calcium, no magnesium, no phenol red (Life Technologies, Carlsbad, CA), 0.5 mM EGTA, 0.5 mM EDTA, 25 mM HEPES, Penicillin-Streptomycin (10,000 U/mL) (Life Technologies) and 0.2% BSA (Research Products International, Mt. Prospect, IL); Liver Digest Media (50 mL for 25g mouse): HBSS, calcium, magnesium, no phenol red (Life Technologies), 0.25 mM HEPES, Penicillin-Streptomycin (10,000 U/mL) (Life Technologies), 3.6 mg Trypsin Inhibitor (Sigma, St. Louis, MO), 28 mg Collagenase Type IV (Life Technologies) added fresh. Digest flow rates were 5 mL/min for 5 min for perfusion and 10 min for digest. The liver was then quickly dispersed and filtered through a sterile 100 μm mesh. Hepatocyte suspensions were then centrifuged at 50x g for 3 min and resuspended to a density of $5 \times 10^6$ cells/ml in DMEM. Viable hepatocytes in the pellet were washed three times and then plated on collagen-coated tissue culture plates in DMEM with 10% calf serum and 100 μg/ml penicillin and streptomycin. After overnight culture, the medium was replaced with F-12 medium containing insulin (10 μg/ml), dexamethasone (67 ng/ml), triiodothyronine (67.3 ng/ml), penicillin (100 units/ml), and streptomycin (0.1 mg/ml). Primary hepatocytes were treated with 5 μg/mL of TM for 4 hr before addition of ActinomycinD (ActD; Sigma) to 5 μg/mL. Cells were collected in triplicate 15 min after addition (considered time 0), 2 hr 15m, and 4 hr 15m after treatment. As all in vitro experiments involved primary cells, cells were not mycoplasma tested.

## Adenovirus experiments

Ad-BiP was created as described (*Young et al., 2012*) and amplified by the University of Iowa Gene Transfer Vector Core. Control virus expressing GFP only (ad-GFP) was also obtained from the Vector Core. $3 \times 10^9$ pfu/mouse were administered through tail vein injections for hepatic expression. Experiments were performed five days later.

## Molecular analysis

RNA and protein analyses were performed as described (*Rutkowski et al., 2006*). Immunoblots were imaged using the ChemiDoc-It imaging system (UVP, LLC, Upland, CA) with on-chip integration and empirically derived exposure settings. Membranes were sliced into appropriate molecular weight ranges for blotting; membranes were never stripped and reprobed. Images were processed using Adobe Photoshop. Black hairlines are solely to aid in visual assessment. All contrast adjustments were performed uniformly. Primer sequences and methods utilized for real-time PCR analysis have been published previously (*Rutkowski et al., 2006*, *2008*; *Arensdorf et al., 2013a*). *Xbp1* RT-PCR was as described (*Tyra et al., 2012*). Additional primers are described here: *Xbp1(s)*: Fwd GAG TCCGCAGCAGGTG, Rev GTGTCAGAGTCCATGGGA; genomic *Hspa5* Primer (5' UTR of *Hspa5* gene) Fwd TAAGACTCCTGCCTGACTGC, Rev GGAATAGGTGGTCCCCAAG; ChIP primers: *Hspa5* Promoter (−215 to +8 bp) Fwd CATTGGTGGCCGTTAAGAATGACCAG, Rev AGTA TCGAGCGCGCCGTCGC; *Hspa5* Intron 7 (+2727 to +2906) Fwd GGGAGGACTGTTGCTTTAGG, Rev TGAATGAACTCTTGCCATCTTC.

## Chromatin immunoprecipitation

ChIP was performed as in (*Arensdorf and Rutkowski, 2013*; *Chikka et al., 2013*) with minor modifications. Formaldehyde-fixed liver tissues were weighed out after being quenched with 1.5M Tris-HCl and pulverized in a metal mortar and pestle. Samples were sonicated using a Covaris E220 sonicator.

Chromatin was immunoprecipitated using ATF6α (sc-22799x, Santa Cruz; RRID:AB_2242950) or Pol II (sc-899x, Santa Cruz; RRID:AB_632359) antibodies or non-specific IgG (12–370, Millipore; RRID: AB_145841) overnight at 4°C. DNA was purified using a standard phenol/chloroform extraction protocol. Samples were then analyzed by quantitative real-time PCR with an annealing temperature of 58°C.

## Rigor and reproducibility

Groups were compared by one-way ANOVA. All quantitative data are presented as means ± S.E.M. Tukey's Post-Hoc analysis was used when comparing multiple conditions for the same readout. Statistical comparisons for qRT-PCR and qPCR data were carried out prior to transforming data out of the log scale. All replicates were biological rather than technical, and 'n' numbers are given in figure legends; for animal experiments, this refers to number of animals. For cell culture experiments, this refers to numbers of independently-treated plates. Each experiment was performed at least twice, and some experiments (such as chronic stress treatments) were performed many times. There was no removal of outliers; no data were discarded unless there was an unambiguous indication that the experiment failed for technical reasons.

## Acknowledgements

We thank R Davisson (Cornell) for the Ad-BiP virus (aka AdGRP78), R Kaufman (Sanford Burnham) for *Atf6-/-* mice and ATF6α antibody, and R Hegde (MRC, Cambridge) for TRAPα antibody. At the University of Iowa, we thank A Peluzzo and J Zabner for *Lep^ob/ob* and *Lepr^db/db* mice, F Jo and C Sigmund for advice on TUDCA experiments; H Qi, K Nillson and D Price for advice on ChIP; and L Yang for stimulating discussions. This work was funded by NIH grant GM115424 to DTR and by funds from the University of Iowa Department of Anatomy and Cell Biology, the Carver College of Medicine, and the Office of the Vice President for Research. J A G was supported by NIH training grant GM067795 as well as funds from the University of Iowa Office of Cultural Affairs and Diversity Initiatives and the Graduate College.

## Additional information

### Funding

| Funder | Grant reference number | Author |
| --- | --- | --- |
| National Institute of General Medical Sciences | GM115424 | D Thomas Rutkowski |
| National Institute of General Medical Sciences | GM067795 | Javier A Gomez |

The funders had no role in study design, data collection and interpretation, or the decision to submit the work for publication.

### Author contributions

JAG, DTR, Conception and design, Acquisition of data, Analysis and interpretation of data, Drafting or revising the article

### Author ORCIDs

D Thomas Rutkowski, http://orcid.org/0000-0001-6586-4449

### Ethics

Animal experimentation: All protocols for animal use, including euthanasia and miminzation of distress, were reviewed and approved by the University Committee on Use and Care of Animals at the University of Iowa, protocol number 4061076.

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
