## [Decision Letter]

Thank you for submitting your article "Experimental Reconstitution of Chronic ER Stress in the Liver Reveals Feedback Suppression of Bip mRNA Expression" for consideration by *eLife*. Your article has been reviewed by three peer reviewers, including Erik Snapp and a member of our Board of Reviewing Editors, and the evaluation has been overseen by Randy Schekman as the Senior Editor.

The reviewers have discussed the reviews with one another and the Reviewing Editor has drafted this decision to help you prepare a revised submission.

Summary:

This study has aimed to design and characterize an experimental paradigm for chronic low-level ER stress in mammals. The reason this is of importance is because many disease states have been correlated with chronic ER stress, but the mechanistic and causal relationships between the stress and disease progression have been challenging to dissect. Characterization of an experimentally tractable system is expected to provide the tools and insights to examine the more complex disease situations. The authors use daily low-dose tunicamycin injections as the paradigm, following up their earlier work in cultured cells with a similar strategy. Analysis of liver at different times by a number of methods (histology, stress pathway activation status, transcriptional status, promoter occupancy, etc.) provides a description of what is happening. The primary discovery is that in the chronic state, levels of mRNA for BiP (and some other UPR targets) are suppressed despite the fact that BiP protein is high and the UPR continues to be responsive to each new dose of tunicamycin. Additional experiments indicate that a signature feature of the chronic stress state is a UPR that is comparably responsive, but more rapidly deactivated. It is proposed that the more rapid deactivation is due to the high levels of BiP protein suppressing UPR sensor activity, and that the low levels of BiP mRNA are due to its degradation, possibly by RIDD. Using this information, the authors provide evidence that a model of obesity (which is associated with low-level ER stress in the liver) shows similar signatures.

Overall, the study's strengths are that it is among the first to try to characterize the chronic ER stress state in vivo, and it appears to have discovered a new homeostatic 'set-point' in which the UPR behaves differently in terms of its kinetics of deactivation. The link to the obesity model is an intriguing observation that appears to have justified their approach.

Essential revisions:

After discussion of the merits and weaknesses of this study, the referees felt that some of the conclusions were not yet fully justified or entirely convincing. In some cases, this could be handled with moderating the claims, while others might require additional experimentation. The following points would need to be addressed.

1) The central concern, voiced by all three referees, was that the causal connections between increased BiP protein, changes in BiP mRNA, and increased RIDD are not established and would require extensive further study. Conclusions to this effect should be tempered, and limited to the Discussion section where it is can be clearly indicated as speculation. It would be more appropriate to simply indicate that, based on *Bloc1s* analysis, there would seem to be evidence for increased RIDD, but that this conclusion and its potential connection to BiP protein and BiP mRNA remains to be investigated in future mechanistic studies.

2) One of the key results of a new set-point is in Figure 4, where the kinetics of *Xbp1* deactivation in chronically stressed liver are faster. Not all the referees were convinced by this experiment, and it would be reassuring to provide an indicator of its reproducibility and robustness. Are the three samples shown at each time point completely independent experimental samples, and has the entire experiment been reproduced with comparable results?

3) Some figures do not compare equivalent timepoints: in the early (acute) studies, 8hr endpoints are compared to 24hr timepoints later (i.e., at D5 and D15). If equivalent timepoint samples are available, it would be informative to know what the chronic state (e.g., D5 or D15) looks like at 8 hrs, and what the acute state (e.g., D1) is like at 24 hrs. This would address questions such as the vacuolization status or changes in ADRP on the later days at the 8 hr time points (in Figure 1) or changes in transcriptional status (in Figure 3). In other words, are the acute changes resolving at comparative timepoints, or are they different?

---

## [Author Response]

*Essential revisions:*

*After discussion of the merits and weaknesses of this study, the referees felt that some of the conclusions were not yet fully justified or entirely convincing. In some cases, this could be handled with moderating the claims, while others might require additional experimentation. The following points would need to be addressed.*

*1) The central concern, voiced by all three referees, was that the causal connections between increased BiP protein, changes in BiP mRNA, and increased RIDD are not established and would require extensive further study. Conclusions to this effect should be tempered, and limited to the Discussion section where it is can be clearly indicated as speculation. It would be more appropriate to simply indicate that, based on Bloc1s analysis, there would seem to be evidence for increased RIDD, but that this conclusion and its potential connection to BiP protein and BiP mRNA remains to be investigated in future mechanistic studies.*

We agree that the data presented are not sufficient to conclusively demonstrate increased RIDD in the chronic condition, and thank the team for recognizing the substantial amount of work that would be required to address this point experimentally. We have addressed this concern by making more explicit in the description of Figure 4 in particular that RIDD might be the active mechanism, but also that further work—in particular, replacing the endogenous IRE1 allele with one for which the *Xbp1* splicing and RIDD activities could be separated and independently modulated—would be necessary to test this model. The Discussion section also contains a fairly lengthy elaboration on this point.

*2) One of the key results of a new set-point is in Figure 4, where the kinetics of Xbp1 deactivation in chronically stressed liver are faster. Not all the referees were convinced by this experiment, and it would be reassuring to provide an indicator of its reproducibility and robustness. Are the three samples shown at each time point completely independent experimental samples, and has the entire experiment been reproduced with comparable results?*

The *Xbp1* experiment shown in Figure 4, as with all other experiments in the paper, represent multiple animals treated independently. Each lane thus represents a separate animal. We have added this notation to the figure legend. In addition, as with all other experiments, it has been repeated to assure reproducibility. The data from a similar but completely separate experiment are shown in Figure 8.

Author response image 1.**DOI:**
http://dx.doi.org/10.7554/eLife.20390.016

This experiment shows both the conventional *Xbp1* splicing assay and data derived from qRT-PCR quantification of both spliced and total *Xbp1* mRNA. By both measures, *Xbp1* splicing in the 5D-treated animals has returned to baseline by 14hrs, but not in the 1D-treated animals. (Note that there is also a background band visible in some samples that is marked by a white circle; it is not spliced Xbp1, as it migrates more slowly and diffusely; spliced *Xbp1* can only be seen in all three D1-8h samples, the first three D1-14h samples, the third D1-24h sample, and all four D5-8h samples.)

Beyond this specific point, we infer that the crux of this criticism is that the original manuscript did not present in convincing enough detail data to support the idea that an essential difference between the chronic condition and the acute is that canonical UPR signaling is deactivated more rapidly in the former than the latter. The additional independent data in Figure 8, as well as the observation that this same behavior is observed for *Bip* mRNA expression (Figure 4) and *Grp94* (not shown), lends confidence to our interpretation that the UPR shuts off more rapidly in the chronic state. In addition, new histological images from D1 and D5 timepoints, included in Figure 1 and Figure 4, further support the idea that the UPR remains responsive but deactivates more rapidly (see below). We have also replaced the original *Bloc1s1* data (Figure 4) with a more complete time course to illustrate the contrast between UPR-dependent transcription—which is deactivated more rapidly in the chronic state—and RIDD activity—which appears by the *Bloc1s1* measure at least to be more persistent in the chronic state. We hope that, together, these data address the substance of this criticism.

*3) Some figures do not compare equivalent timepoints: in the early (acute) studies, 8hr endpoints are compared to 24hr timepoints later (i.e., at D5 and D15). If equivalent timepoint samples are available, it would be informative to know what the chronic state (e.g., D5 or D15) looks like at 8 hrs, and what the acute state (e.g., D1) is like at 24 hrs. This would address questions such as the vacuolization status or changes in ADRP on the later days at the 8 hr time points (in Figure 1) or changes in transcriptional status (in Figure 3). In other words, are the acute changes resolving at comparative timepoints, or are they different?*

We agree that this was an (unintentionally) confusing aspect of the original manuscript and have revised the data to clarify, particularly in Figure 1. Now this Figure shows D1-8h treatments separately (Figure 1), and only to illustrate the capacity of 0.025 mg/kg TM to activate the UPR acutely. It now also compares D1-24h and D5-24h treatments by both histological and immunoblot analysis. This is the more relevant comparison and substantiates our point that the chronic condition is qualitatively distinct from the acute condition—i.e., that the chronic condition is not simply an artifact of harvesting at 24h rather than 8h. We have also replaced the ADRP immunostaining shown in original Figure 1 with an ADRP immunoblot, which unambiguously describes the state of lipid accumulation at the D1 and D5 periods (and was necessary due to unforeseen technical issues with ADRP immunostaining that we have been unable to resolve). Together, both the histology and ADRP immunoblotting demonstrate substantial lipid accumulation at D1-24h, which has largely resolved by D5-24h. In addition, we have also added H&E staining from the D1-8h and D5-8h timepoints to Figure 4, which, along with the H&E data in Figure 1, show that lipid dysregulation occurs in the chronic condition but is resolved more rapidly.

For the ChIP experiments shown in Figure 3, the 8h time point is used only as a positive control, to confirm that we can detect ATF6α or Pol2 binding when it is occurring. We can infer that ATF6α binding to the *Bip* promoter at the D5-8h time point must be robust simply because the *Bip* transcript level is elevated to a comparable degree as at D1-8h (Figure 4). Likewise, we can infer that, even in the D1 condition, ATF6α binding is reduced to or very near to basal levels by 24h simply because of the diminishment of *Bip* mRNA expression even on the first day. However, the ChIP experiments do not offer enough resolution to compare ATF6α (or Pol2) binding between the D1-24h and D5-24h conditions—i.e., to conclude that the binding at D5-24h is less than the binding at D1-24h, since both are at or near basal levels. We have altered the text describing Figure 3 to make clearer the goals of these experiments.